# Differences and Similarities in Protein and Nucleic Acid Structures and Their Biological Interactions

**DOI:** 10.3390/cimb47121019

**Published:** 2025-12-06

**Authors:** Tsutomu Arakawa, Taiji Oyama, Tomoto Ura, Suguru Nishinami, Kentaro Shiraki, Teruo Akuta

**Affiliations:** 1Alliance Protein Laboratories, 13380 Pantera Road, San Diego, CA 92130, USA; 2Sales Division, JASCO Corporation, 2967-5 Ishikawa, Hachioji 192-8537, Japan; taiji.oyama@jasco.co.jp; 3Institute of Pure and Applied Sciences, University of Tsukuba, Tsukuba 305-8573, Japan; ura.tomoto.gn@u.tsukuba.ac.jp (T.U.); shiraki@bk.tsukuba.ac.jp (K.S.); 4Institute for Genetic Medicine, Hokkaido University, Sapporo 060-0815, Japan; nishinami@igm.hokudai.ac.jp; 5Research and Development Division, Kyokuto Pharmaceutical Industrial Co., Ltd., 3333-26, Aza-Asayama, Kamitezuna, Tahahagi 318-004, Japan; t.akuta@kyokutoseiyaku.co.jp

**Keywords:** peptide backbone, side chain, phosphate backbone, aromatic side chain, secondary structure, molecular interactions

## Abstract

Protein and nucleic acid play central roles in biology and pharmaceuticals. Both share a similar architecture made of a backbone and side chains. Protein has a peptide backbone and various side chains, whereas nucleic acid has a phosphate backbone and aromatic side chains. However, they are significantly different in the chemical properties of the backbone and side chains. The protein backbone is uncharged, while nucleic acid backbone is negatively charged. The protein side chains comprise widely different chemical properties. On the other hand, the nucleic acid side chains comprise a uniform chemical property of aromatic bases. Such differences lead to fundamentally different folding, molecular interactions and co-solvent interactions, which are the focus of this review. In regular protein secondary structures, the peptide groups form polar hydrogen bonds, making the interior hydrophilic. The side chains of different chemical properties are exposed on the outside of the protein secondary structures and participate in molecular and co-solvent interactions. On the other hand, hydrophobic/aromatic nucleobase side chains are located inside the typical double helix or quadruplex structures. The charged phosphate groups of the nucleic acid backbone are located outside, participating in electrostatic interactions. The nucleobases are also involved in molecular interactions, when exposed in breaks, hairpins, kinks and loops. These structural differences between protein and nucleic acid confer different interactions with commonly used co-solvents, such as denaturants, organic solvents and polymers.

## 1. Introduction

Protein and nucleic acid are key players in biological systems and pharmaceuticals. Proteins and nucleic acids are currently two important biological macromolecules in the research and biopharmaceutical sectors [1,2,3,4,5,6,7,8]. They are made of a similar architectural framework of a polymer backbone and side chains, which is folded into a secondary, tertiary and quaternary structure. When folded, proteins confer a variety of functions by binding to target molecules, e.g., substrates for enzymes, cell surface receptors and cellular signaling factors for signal transduction and nucleic acids for the regulation of transcription and translation. Such bindings are in large part mediated by amino acid side chains (e.g., charged, hydrophilic, hydrophobic and aromatic), which are exposed to the protein surface of the folded structure. Peptide groups, when exposed, can mostly participate in hydrogen bonding to form regular secondary structures of α-helix and β-sheet. On the other hand, the side chains of nucleic acid are in large part buried in double or other multimeric helices and thus do not participate in molecular interactions, except for the event of transcription and translation, or when exposed, as in breaks, loops, hairpins and kinks of nucleic acids; however, the side chains do participate in molecular interactions, e.g., binding to proteins. The charged backbone of nucleic acids that are generally exposed play a major role in interacting with proteins that stabilize and regulate the folding and function of nucleic acids. This review focuses on the common structural framework shared by protein and nucleic acid and on the differences in folding pattern between them, leading to different molecular interactions and interactions with commonly used co-solvents.

## 2. Structure

Figure 1 shows the primary structure of protein and nucleic acid. Both structures are essentially made of a backbone and side chains. Protein is composed of peptide backbone groups flanked by carbon atoms, to which side chains (R) are attached. Similarly, nucleic acid is composed of phosphodiester linked backbone flanked by ribose, to which nucleobases (B) are attached. As is evident, protein and nucleic acid have a common architecture, a backbone and side chains. However, there are two critical differences between protein and nucleic acid. One is the charges on the backbone, and another is the chemistry of side chains.

The backbone of protein consisting of peptide groups bears no charges, meaning that the peptide groups can be easily accommodated into a folded structure without electrostatic repulsive interactions. The peptide groups form hydrogen bonds and are buried in the regular secondary structures of α-helix (blue) and β-sheet (green), although some peptide groups are exposed in the extended, disordered structures, as depicted in Figure 2. The interior of the secondary structures is hydrophilic and segregated from the environments created by side chains or solvents. Such an incorporation of backbone into the secondary structure cannot occur with nucleic acids, in which the backbone phosphate groups bare negative charges and normally are exposed to interact with counter ions or other oppositely charged groups.

Side chains are also significantly different between protein and nucleic acid. The side chains of protein are composed of widely different chemical properties, as schematically shown by R1, R2, R3, R4, etc., in Figure 1. Tryptophan (Trp), tyrosine (Tyr) and phenylalanine (Phe) are aromatic and hydrophobic and participate in hydrophobic and aromatic interactions as in π–π stacking and π–cation interactions. Leucine (Leu), isoleucine (Ile), valine (Val) and alanine (Ala) are hydrophobic, while threonine (Thr), serine (Ser), glutamine (Gln) and asparagine (Asn) are hydrophilic and are normally exposed to solvents in the folded tertiary structure. Aspartic acid (Asp) and glutamic acid (Glu) are negatively charged and can contribute to electrostatic ion pairs and anion–π interactions [9]. Arginine (Arg), lysine (Lys) and histidine (His) carry a positive charge and can participate in electrostatic interactions with ions or oppositely charged groups. These positive charges, particularly Arg, participate in key molecular interactions with aromatic compounds and aromatic side chains as π–cation interactions, as described later [10,11,12]. Arg has a planer side chain structure that can form π–π stacking with aromatic groups [13,14,15].

Nucleic acid is different in the chemical property of side chains from proteins. The side chains of nucleic acid are made of a highly similar chemical property. Namely, they are aromatic and hydrophobic, as is evident from their favorable (attractive) interaction with organic solvents, alkyl-urea and alkyl-amides, particularly with longer alkyl-chains [16,17]. The aromatic properties of nucleobases are also evident from their attractive interactions with the aromatic arginine side chain, i.e., guanidinium group [18]. Thus, when protein and nucleic acid are folded into a secondary structure, the principal interaction mechanism is fundamentally different between them.

Major secondary structures of protein are α-helix and β-sheet, as depicted in Figure 2. Key factors in these protein structures are as follows. The forces that stabilize them are hydrogen bonding between peptide groups, i.e., no involvement of side chains. Thus, the inside of helix and sheet is hydrophilic, comprising hydrogen bonds. As described later, such hydrophilic hydrogen bonding is augmented in non-polar environments provided by hydrophobic side chains, organic solvents or the binding of amphiphilic detergents [19,20,21,22,23,24]. The outside of these regular secondary structures of protein is made of side chains with widely different surface chemistries, as depicted in Figure 1 and Figure 2. Depending on which side chains are exposed, the surface of the protein’s secondary structures can be charged, hydrophilic, hydrophobic or aromatic and can participate in a variety of interactions responsible for the formation of tertiary structure folding [9] or interactions with other molecules for the formation of quaternary structures [25]. It can lead to self-association in oligomeric or assembled structures, such as oligomeric enzymes, collagen, F-actins and microtubules [26]. Uneven distribution of these side chains can generate specificity in molecular interactions.

One such example of uneven side chain distribution is called “helical wheel”, schematically shown in Figure 3 [27,28,29]. In α-helical structures, hydrophobic side chains are often distributed in one side of a helix (red line), while the other side is made of hydrophilic side chains (blue). This would make one side of the helix too hydrophobic to be exposed to aqueous environments and hence it will tend to be buried in hydrophobic environments. This can be achieved by interacting with other helices with similar surface properties (2 red lines), when another helical wheel structure is available, as seen in Figure 3, forming so-called α-helical bundles (right panel). This plays two important roles. Hydrophobic interaction between two or more helices mutually stabilizes both, as a single helix is not stable but can be stabilized by bundling or burying the helix into hydrophobic microenvironments, which destabilizes the exposure of polar peptide groups. Exposure of peptide bonds to water may cause the dissociation of inter-peptide hydrogen bonding and instead form hydrogen bonds with water molecules [30,31], namely, hydrogen bonds between peptide groups are more stable in hydrophobic environments. Another important factor in helical wheel and α-helix bundling is the hydrophilicity of the other surface (two blue lines), which would make the outside surface water-soluble. Namely, the overall folded structure becomes water soluble by burying non-polar groups inside.

A typical α-helix (blue) and β-sheet (green) structure is stabilized by hydrogen bonding between peptide groups, which render the interior of these structures hydrophilic. The exterior surface of α-helix and β-sheet are composed of amino acid side chains of different chemical properties. The disordered structures have both peptide groups and side chains exposed. Molecular interactions between these structures make the protein fold into a distinct native globular structure.

A helix often assumes an amino acid orientation called “helical wheel”, in which one side of the α-helix has a higher population of hydrophobic side chains and the other side has a higher population of hydrophilic side chains. Two or more α-helices bind together through a hydrophobic surface, which would stabilize the helix. The other hydrophilic surfaces are exposed to aqueous solvent, which would make the helices more water-soluble.

This cannot happen in nucleic acid having very different backbones and side chain chemistries compared to those of protein. However, a protein-like secondary structure may be possible by attaching nucleobases to carbon atoms, which are flanked by peptide groups. Such a construct is called “nucleopeptides”, which are composed of nucleobases inserted on a peptide backbone [32]. They exhibit interesting features due to their capacity to bind complementary single-stranded RNA and DNA sequences via hydrogen bonding between a nucleopeptide and a single-stranded nucleic acid. As in a nucleic acid double helix or quadruplex, base stacking in the nucleopeptide and the single nucleic acid strand can enforce hydrogen bonding. There appears to be no evidence of protein-like α-helix or β-sheet in nucleopeptides by CD measurements; namely, the nucleopeptide does not appear to form peptide group-mediated hydrogen bonds and the resultant protein-like helix or sheet.

What happens in nucleic acid structure is shown in Figure 4. It shows a typical double helix with nucleobase and ribose groups inside and the negative charges of phosphodiester linkage outside. Its inside is thus occupied by hydrophobic/aromatic nucleobases forming aromatic stackings and hydrogen bonds between the base pairs. Such a structure renders the inside aromatic and hydrophobic and, when exposed, can be stabilized by non-polar organic solvents, different from peptide hydrogen bonds that will be destabilized by non-polar organic solvents. Because of bulky bases inside the nucleic acid helix, the distance between two strands is 2 nm (see Figure 4), larger than the 1.2 nm (Figure 2) of protein helix diameter. It is important to note that the outer surface of the nucleic acid double helix is very different from the outer surface of the protein helix. The outer surface of a nucleic acid double helix or quadruplex is highly charged and hydrophilic, while the protein’s helix has the outer surface composed of variable chemistry, i.e., charged, hydrophilic, hydrophobic or aromatic.

Thus, the formation of packed tertiary structure folding is very different between protein and nucleic acid. As described earlier, side chains of protein secondary structures are fully exposed and can interact with other side chains. Such side chain-mediated interactions lead to the packing of the secondary structures into a folded tertiary structure with a resultant globular shape. As shown in Figure 2, the exposed peptide groups as well as the side chains of the disordered secondary structure can also participate in the formation of the folded structure. There are many key side chains in the packed structure. Among them, the charge pair between negatively and positively charged side chains is buried and stabilized by mutually neutralizing the charges. Such electrostatic interaction could be stronger in the hydrophobic environment of the protein’s interior due to the low dielectric constant [33]. Hydrophobic side chains are segregated from solvent water and stabilized by interacting with other hydrophobic side chains and form a core structure. In addition, aromatic groups also play a key role in the folded structure by π–π interaction with other aromatic side chains and π–cation interaction with positively charged side chains, particularly arginine side chain, which have a planer structure, as has been described earlier.

These interactions in protein’s packed structure do not occur in nucleic acids. As described earlier, side chains of nucleic acid are buried inside of the double helix or quadruplex and the outside surface has a constellation of negative charges, markedly different from the chemistry of the outer surface of protein’s helix or sheet. Thus, the folding of nucleic acid, if it occurs, must accompany charge neutralization by counter ions to suppress the charge repulsion of backbone phosphate groups or binding to positively charged side chains from other compounds, e.g., polyamines, or proteins. Such nucleobase side chain-mediated interaction may occur through terminal bases, which are at least partially solvent-exposed. In addition, the double strand helix of nucleic acid often has breaks, kinks, loops and hairpins, as depicted in Figure 5, which result in exposure to nucleobases. The flipping of nucleobases can also occur and exposes those nucleobases to interaction with other structures.

## 3. Interactions

Amino acid side chains play a key role in not only the folding of tertiary structure but also inter-molecular interactions, whether specific or non-specific. Proteins perform specific functions by binding to themselves or target proteins or nucleic acids or undergoing non-specific aggregation. Nucleic acids also perform functions by binding to proteins. These molecular interactions are in vitro controlled by so-called “co-solvents” including salts, sugars, polyols and polymers, from which we can learn about the nature of molecular interactions.

### 3.1. Macromolecular Interactions

Amino acid side chains provide specific interactions within a protein molecule to generate a unique three-dimensional structure that is critical for their functions. In the folded structure, protein molecules have the solvent-exposed side chains of different chemical properties, as described earlier, which participate in protein–protein interactions. Figure 6 shows such examples. Figure 6A shows two folded proteins (human G-protein subunit alpha; G_αi3_ and engineered regulator of G-protein signaling type 2 domain; RGS2) that form a specific complex, a critical protein–protein complex for signal transduction [34]. First, it is noted that both proteins are packed into a more or less globular shape, which generates a rigid surface structure. These two proteins bind to each other at a specific site, as seen in Figure 6, with a tightly fit interface, with complementary morphology. Many side chains, e.g., Glu65, Thr182 and Lys210 of G_αi3_ and Ser106, Lys191, Asp184 and Arg188 of RGS2, are involved in conferring electrostatic interactions and hydrogen bonding, which generate the specific and strong binding of these two proteins, as shown in Figure 6A.

Figure 6B shows the antigen–antibody interaction in which a human antibody, durvalumab Fab domain, binds to an antigen, PD-L1 [35]. It is shown here that both the antibody Fab and PD-L1 domains are also more or less folded into a rigid tertiary structure. As shown, multiple side-chain contacts contribute to the interface, including bidentate salt bridges between PD-L1 Arg113 and the antibody heavy-chain Asp31, and an ion pair between PD-L1 Glu58 and light-chain residue Lys52. In addition, PD-L1 Asp26 forms an ion pair with the antibody light-chain Arg28, accompanied by several hydrophobic contacts. Notably, PD-L1 Arg125 forms an unusual hydrogen bond with the antibody peptide backbone. In the case of atezolizumab n Figure 6B), Tyr56 of PD-L1 engages in hydrophobic/π–π interactions with heavy-chain Trp33, Trp50 and Ser57. Among them, aromatic contacts are thought to play a central role in mediating strong and specific antigen–antibody binding [36,37,38]

Protein–nucleic acid interactions are not only fundamental to biological regulation and structural stabilization but are also increasingly exploited in therapeutic and diagnostic developments. Such advances as CRISPR-related genome-editing technologies [39,40], protein-binding RNA aptamers [41] and mRNA vaccines that rely on optimized RNA–protein interactions for translation and immune activation [42] exemplify how these interactions have been harnessed for medical applications. These and other modalities underscore the growing importance of understanding protein–nucleic acid interfaces at atomic resolution for rational drug and therapeutic design. Figure 7 shows a few examples of protein–nucleic acid interactions that play a central role in regulating and stabilizing nucleic acids. Here, a few RNA-binding proteins with three different structures of RNA are shown. The global folding of proteins and nucleic acids appears to be different. As in Figure 6, these proteins are folded as a globular shape, while nucleic acids lack such a rigid folding, which in turn suggests a more flexible nature of nucleic acid folding than protein folding. Figure 7A shows the binding of splicing factor 1 (SF1) with a single-stranded RNA to generate a spliceosome assembly [43]. In the SF1–RNA complex (PDB 1K1G), the 5′-terminal bases (UAC) are stabilized by the QUA2 helix (residues 239–249), while the 3′-terminal bases (UAAC) interact with the KH domain, particularly residues P159, R160, and G161 within the Gly–Pro–Arg–Gly loop, via hydrogen bonds and hydrophobic contacts. Lys and Arg in SF1 bind to the RNA backbone phosphate groups by electrostatic interactions. One of the exposed bases is deeply buried in a hydrophobic cleft comprising a Gly–Pro–Arg–Gly motif and hydrophobic (Leu, Ile, Val)/aromatic (Tyr, Phe) amino acid residues. As summarized in Figure 7A, a few solvent-exposed aromatic nucleobases as well as phosphate backbones interact with the solvent-exposed amino acid side chains through electrostatic interactions, hydrogen bonding and hydrophobic/aromatic interactions. Figure 7B shows a case of double-stranded RNA, in which an RNA-cleaving enzyme, ADAR2 deaminase, binds to a DNA double helix [44]. Glue488 of the deaminase intercalates into the RNA double helix and binds to the orphan (unpaired) base by hydrogen bond and enables the base to flip into the enzyme’s active site. The enzyme’s residue 454–477 that includes Arg455 forms a loop that inserts into the target RNA and contacts the phosphate backbone. Such RNA-binding renders the enzyme’s loop ordered in a structure and causes the base to flip. Multiple basic residues, e.g., Lys594 and those in the loop, form electrostatic interactions with the phosphate backbone, as seen in Figure 7B.

Figure 7C shows an example of the binding of a folded RNA to an RNA-binding protein L7Ae, a typical model of a K-turn binding motif [45,46]. The L7Ae binds to the K-turn structure formed by two sheared base pairs and stabilizes the kinked RNA conformation, as seen in Figure 7C, using various amino acid side chains. The Glu38 forms hydrogen bonds with the terminal guanine base. Arg95 interacts with the phosphate backbone and Ser94 hydrogen bonds with the RNA backbone. Val94 and Ile92 interact with the terminal guanine by hydrophobic interactions, as summarized in Figure 7C.

Another example of protein–nucleic acid interaction is described in Figure 8 for quadruplex DNA, which is formed by a single-stranded DNA, in which four bases stack over each other [47]. Three major binding sites on the quadruplex structure have been demonstrated, i.e., top-stacking [48], groove-binding [49] and loop-binding [50]. In top stacking, it has been shown that Arg and Lys of the quadruplex-binding protein DHX36 bind to phosphate backbone via electrostatic and hydrogen bond interactions, while the aromatic amino acid of DHX36 has hydrophobic interaction or aromatic stacking with the terminal guanine base. In groove-binding, the telomeric end-binding protein binds to the quadruplex groove by Arg/Lys participating in electrostatic interactions with the phosphate backbone and Tyr participating in aromatic stacking with the guanine base. In loop-binding, e.g., binding of zinc finger proteins, a basic His/Arg/Arg cluster binds to the outward-directed nucleotides, most likely through cation–π or π–π interactions.

### 3.2. Specific Role of Arginine in Molecular Interaction

In protein–protein and protein–nucleic acid interactions, arginine side chains can play a central role by engaging in various molecular interactions, including electrostatic ones with charged groups and aromatic interactions with aromatic side chains. The mode of interactions of arginine side chains with other side chains has been well documented in protein structure. Among them, cation–π and π–π interactions play a significant role in protein folding and protein–protein interactions, as described above. How does arginine contribute to the interactions of protein with nucleic acid? Different from proteins, nucleic acids are highly charged, and hence electrostatic force makes a larger contribution to the interaction of the basic arginine side chain with nucleic acids than with the proteins. It has been shown that arginine residues participate in hydrogen bonding to the phosphate groups in nucleic acids [51]. This hydrogen bonding was often termed “arginine fork”, as the arginine side chain can form a specific network of hydrogen bonds with phosphate groups of RNA. In virus particles, arginine side chains (more likely just a guanidinium group of Arg side chain) bridge with phosphate groups in the virus nucleic acids, while the aromatic side chains of the viral proteins engage in hydrophobic interactions with nucleobases, and the aspartic acid side chains establish hydrogen bonds with sugar groups of RNA within the viral structure [52,53]. Arginine forks can participate in multiple hydrogen bonding with phosphate groups and nucleobases to mediate nucleic acid binding to proteins [51,54,55].

The participation of hydrogen bonding between the arginine side chain and phosphate groups has also been indicated in not only protein–nucleic acid interaction but also protein–protein interaction via phosphorylated serine residues on the proteins [56]. Such interaction of arginine side chains was also observed with polyphosphate compounds via bivalent hydrogen bonding, which cannot be achieved by lysine residues [57]. Arginine residues interact with nucleic acids via hydrogen bonding to phosphate backbone groups and nucleobases, and also via π–cation interaction with the nucleobases [58]. The ability of arginine to participate in multiple binding mechanisms for nucleic acids can lead to the formation of stable protein–nucleic acid complexes. One such binding mechanism is the aromatic/π–cation interactions of the arginine side chain with nucleobases.

Solubility measurement has been used to demonstrate interactions of nucleobases with amino acids and related compounds [18]. The above study showed a monotone increase in solubility with increasing arginine and GdnHCl concentration. Both arginine and GdnHCl co-solvents were effective in increasing the solubility of these bases, while glycine and NaCl either were ineffective or even decreased the solubility [18]. The results indicate that the interaction between arginine and nucleobases is not of an electrostatic nature, as the electrostatic interactions can be replaced by NaCl and glycine. Between GdnHCl and arginine, arginine showed more favorable, attractive interactions with nucleobases. Arginine at 1 M was most effective on adenine (2.5-fold increase in solubility), followed by cytosine, guanine, thymine and uracil (1.3-fold increase), resulting in the same solubility order for these five nucleobases in 1 M arginine as in the pH 7 buffer. There appears to be no particular correlation with their structure parameters, e.g., the number of double bonds. The above observation is consistent with the reported favorable interaction of arginine residues (i.e., arginine side chain) with nucleic acids, as described above.

### 3.3. Biomolecular Condensates

The molecular complexes described above involve specific and strong interactions between proteins, between nucleic acids and between protein and nucleic acid. There are other types of biomolecular interaction that may occur in cells or in vitro. They are weak and transient molecular interactions, which may be augmented at high macromolecular concentrations (i.e., crowded environments) and often appear in cellular environments, forming a network structure termed “condensate”. When formed in vitro, such a network structure can cause high viscosity, which leads to serious problems in pharmaceutical development, e.g., downstream process of protein purification and parenteral injection [59]. Recent studies have highlighted the crucial role of biomolecular condensates that regulate various cellular functions [60]. Proteins and nucleic acids are the principal molecular drivers of the condensates, and the mechanisms underlying their interactions are gradually being elucidated. In the context of condensate formation in cells under physiological environments, cooperative interactions, although weak and transient, between proteins and nucleic acids are involved. Under controlled in vitro settings, both proteins and nucleic acids are also individually capable of forming network structure, allowing detailed mechanistic analyses of condensate formation.

Condensates that are formed in cells or in vitro are detected by turbidity (light scattering) or hydrodynamic and thermodynamic measurements. Although their interrelation is not fully demonstrated, those condensates are often accompanied by optical observation as liquid droplets, termed “liquid–liquid phase separation”. Due to the difference in refractive index between bulk solvent and condensate, those droplets can be detected by light microscopy. Those membrane-less cellular droplets/organelles include, but are not limited to, nucleolus, speckles, paraspeckles, nuclear bodies, Cajal bodies, P-bodies and stress granules. Such phase separations are also observed in vitro under well-controlled conditions.

Multivalent avidity binding mode between two flexible molecules (A). Multiple interaction mode between four flexible molecules (B). Conversion of non-globular condensate/liquid droplets (C) to globular condensate (D). This is an original diagram drawn by us.

A common feature of both systems is their ability to form multivalent and dynamic interaction networks. Molecules with high structural flexibility and multiple interaction motifs are more prone to condensate formation. It should be noted that the multiple interactions described here differ from the multivalent interactions that confer binding avidity, as depicted in Figure 9. Figure 9A shows multivalent (here trivalent) interactions between two macromolecules, which confer binding avidity. Since these interactions occur on the same two molecules, each interaction adds binding energy to the dimeric assembly, leading to its strong interactions. On the other hand, as shown in Figure 9B, two molecules are bound by only a single interaction. However, each molecule offers multiple binding sites for other molecules, which lead to many molecules being connected to each other, generating a network of macromolecular chains. This generates a network structure throughout the solution, a cause of high viscosity. When the network structure is separated from the bulk phase, it should lead to a higher macromolecular density and refractive index, resulting in the appearance of light scattering and refractive index gradient between the bulk and the solution phase containing the network.

It has been shown that the liquid droplet formed around the condensates are often spherical or globular, not fully expected from the possible irregular shape of condensates. This may be explained by the dynamic and fluid nature of the liquid droplets. Since the condensate creates a higher density and refractive index, it generates an interface between the droplet and bulk and hence the interfacial tension. If the condensate and droplet are not globular, as shown in Figure 9C, the interfacial tension and hence the surface free energy is greater than those expected of the globular shape. When the interior of the droplet is fluid and not viscous, and the molecular interactions stabilizing the condensate are weak, flexible and transient, the network structure trends to rearrange to become more globular, as shown in Figure 9D, thereby reducing the surface free energy. If the droplet phase is viscous and the molecular assembly in the condensates is rigid, the network structure will not rearrange, and the droplet will stay non-globular.

In proteins, intrinsically disordered regions (IDRs) and low-complexity regions (LCRs) provide multiple weak interaction sites—such as the RGG motif [61] and aromatic residues [62]—that drive reversible condensation. Similarly, in nucleic acids, flexible single-stranded RNAs exhibit higher phase-separation propensity than rigid double helices [63]. In this sense, proteins and nucleic acids are similar in terms of the structure that drives the formation of condensates and liquid droplets. Although the negatively charged phosphate backbone may seem to hinder self-assembly, base pairing and base stacking between nucleobases can facilitate condensation [64]. Furthermore, planar higher-order structures such as guanine quadruplexes (G4) can generate hydrophobic π-stacking interfaces that serve as nucleation sites for condensation [65]. In cells, such homotypic interactions are complemented by heterotypic interactions between proteins and nucleic acids, which play a central role in driving condensates. Positively, the solvent-exposed charged side chains of proteins, such as arginine and lysine, electrostatically interact with the negatively charged phosphate backbone, while π–π and π–cation interactions between aromatic and basic residues in proteins and nucleobases exposed in single strand or loops, breaks, hairpins and kinks of double and quadruplex nucleic acids stabilize the condensates [66]. These cooperative interactions give rise to protein–RNA condensates (RNP condensates), forming the basis of the membrane-less cellular structures described above. Higher-order nucleic acid structures—including G4s and hairpins [67]—also function as structural elements that modulate condensate formation. While the importance of higher-order nucleic acid structures in regulating biomolecular condensates has been increasingly recognized, their precise roles in condensate regulation remain an intriguing subject for future investigation.

The RGG motif forms condensates not only with RNA but also with folded DNA sequences, such as the promoter region of the oncogene c-Myc and the G_4_C_2_ repeat of the C9orf72 gene, which has been implicated in amyotrophic lateral sclerosis (ALS) [68,69]. Moreover, not only flexible peptides but also chromatin-associated proteins containing both well-folded domains and intrinsically disordered regions can form condensates with the KRAS promoter G4 [70]. These condensates between nucleic acids and proteins require the formation of a folded G4 structure. A 5-methylcytosine modification of DNA has been shown to influence condensate formation, an effect that arises from its ability to modulate the topology of G4 structures [69]. The formation of G4 creates planar G-quartet surfaces, multiple loops, and flanking regions, which together provide an ideal platform for multiple interactions with proteins [71,72]. Furthermore, a G4 formation in the BCL3 promoter has been shown to promote transcriptional condensate formation, whereas the G4 in the transcript exerts negative feedback on condensate formation, suggesting that DNA and RNA G-quadruplexes mutually influence condensate dynamics [73]. It appears that the G4 structure is unique in that, although folded, it can drive condensate formation.

### 3.4. Co-Solvent Interaction

This review focuses on the similarities and differences in structure between protein and nucleic acid, which result in different folding and molecular interactions. Why then should this review include co-solvent interactions with protein and nucleic acid? Nowadays, co-solvents are widely used in vitro for manipulating aqueous solutions containing protein and nucleic acid, e.g., in purification, chromatography, long-term storage, formulation and solubilization. Understanding how the co-solvents interact with these macromolecules should help design their optimal applications. However, earlier studies on co-solvent interactions with proteins were aimed at understanding the nature of protein folding. Namely, the co-solvent interaction pattern was used to define the hydrophobicity scale of amino acid side chains and peptide backbones, which was then correlated with their preference for water or the internal hydrophobic core [74,75]. Thus, the introduction of earlier studies on co-solvent interaction should shed light on the differences in protein and nucleic acid folding.

#### 3.4.1. Organic Solvent

Acetone and ethanol are among the classic organic solvents used to precipitate proteins and nucleic acids [76,77,78,79]. There is a wealth of information on the interaction between protein side chains and organic solvents [74,80,81,82,83,84]. It was shown that a highly polar glycine, which has a negligible side chain contribution to solvent interaction, drastically loses aqueous solubility with the increasing concentration of organic solvents, including ethanol, dioxane, ethylene glycol and dimethyl sulfoxide, indicating that the interactions are repulsive between the polar glycine and the organic solvents. It is noted that, although glycine has terminal amino and carboxyl groups, it is a constituent of the peptide backbone structure. Hydrophilic groups of protein side chains, e.g., glutamine and asparagine, also have repulsive interactions with the organic solvents. Namely, these hydrophilic side chains and glycines dislike non-polar environments and like aqueous environments. The histidine side chain showed negligible interactions with the organic solvents, which may be ascribed to a balance between the repulsive hydrophilic interaction and attractive aromatic interaction of the histidine side chain with the organic solvents. There are no interaction data available between charged side chains and organic solvents. However, one can speculate from the glycine data that charged amino acid side chains (i.e., Asp, Glu, Lys and Arg) have repulsive (unfavorable) interactions with the organic solvents, consistent with the repulsive interaction of ionic electrolytes with organic solvent [85]. More importantly, polar peptide groups, which constitute the polypeptide backbone structure, showed repulsive interactions with organic solvents, e.g., ethanol, dioxane and ethylene glycol [74,81]. This means that the peptide groups will be segregated from contact with organic solvents and favor inter-peptide interactions or interactions with other polar groups. Nozaki and Tanford also performed extensive measurements of the interactions between the aromatic/hydrophobic groups of protein side chains, i.e., Trp, Tyr, Phe, Ile, Leu, Val and Ala, and organic solvents [74,81], and this has also been performed more recently [86]. Organic solvents showed attractive interactions with these hydrophobic and aromatic groups.

The interactions of nucleic acid constituents with organic solvents have not been studied as extensively as with those of proteins. The mechanism of phosphate backbone interaction with organic solvents should be the same as the mechanism of protein charges, and hence repulsive. The repulsive interactions between organic solvents and phosphate backbones forming charge clusters (see Figure 4) could even be stronger than separated charged amino acid side chains. The question is whether nucleobases are similarly aromatic to the aromatic side chains of proteins. To our knowledge, there seems to be little data for the interactions between aromatic nucleobases and organic solvents, although we may expect them to be attractive, as seen with the attractive interactions between organic solvents and the aromatic side chains of the proteins. Solubility measurements showed a monotone increase in the solubility of adenine in the presence of several organic solvents, including *N*,*N*-dimethylformamide, *N*-methyl pyrrolidone, propylene glycol and dimethyl sulfoxide [87] and uracil in the presence of ethanol below ~50% [88]. Thus, these heteronuclear aromatic nucleobases showed favorable, attractive interactions with non-polar organic solvents, which are like the aromatic amino acid side chains in proteins.

Native protein is unfolded by organic solvent or denaturants, which may form an intermediate structure, which can cause a loss of interactions between regular secondary structures. Denaturants can further unfold the secondary structures, while organic solvent may stabilize the secondary structures. This is an original diagram drawn by us.

How do organic solvents affect protein and nucleic acid structure and what do we learn from the effects of organic solvents on their structure? Figure 10 schematically shows the effects of organic solvents on protein structure. Organic solvents induce denaturation of the proteins, which can be explained by organic solvents binding to (structure-stabilizing) the hydrophobic core comprising mostly hydrophobic amino acid side chains [82,89,90]. This is consistent with the established hydrophobic scale of side chains, as those side chains are segregated from aqueous environments and buried in the interior of the protein molecules. Upon unfolding, these side chains are stabilized by organic solvents by hydrophobic interactions. How do organic solvents affect secondary structures? Two factors need to be considered in understanding the effects of organic solvents on protein secondary structure. The first factor is the consequence of denaturation (the unfolding of tertiary structure), which is brought by the organic solvents. This eliminates interactions between secondary structures, i.e., inter-helical or inter-sheet interactions. For example, it was shown that the hydrophobic side chain interactions of the helical wheel stabilize helices forming bundles, which will be eliminated during denaturation by organic solvents. Thus, this factor should result in the unfolding (loss) of the secondary structure. However, the second factor operates against the unfolding effects of organic solvents on secondary structure. Namely, the interaction between the polar peptide group and the organic solvent is repulsive, which favors the segregation of peptide groups from non-polar environments of organic solvents. Thus, this second factor by organic solvents should stabilize the secondary structure, as shown in Figure 10. Which of two factors dominates the other depends on the non-polar nature of the organic solvents used and the strength of the hydrophobic interactions between the secondary structures. In fact, a number or organic solvents and detergents that provide non-polar environments for peptides have shown an enhanced helical formation [91,92,93,94].

The effects of organic solvents on nucleic acid secondary structure may be more straightforward. The organic solvents most likely melt the secondary structure of the nucleic acid by favorable (attractive) interactions with nucleobases [95,96,97], although hydrogen bonding between base pairs may be destabilized by organic solvents, as in the hydrogen bonding of peptide groups. If melting occurs, then strong electrostatic repulsion between phosphate charges would cause expansion of the nucleic acid structure, as shown as a more extended structure in Figure 11. A lower dielectric constant of organic solvents would augment the charge repulsion. Thus, we can derive from the effects of organic solvents that the protein secondary structure is made of the hydrophilic (organic solvent repulsive) interactions of peptide groups, while the nucleic acid secondary structure is made of the hydrophobic (organic solvent attractive) interactions of nucleobases.

Organic solvents and denaturant destabilize the helix. While organic solvents cause expansion of the dissociated strand, denaturants may allow local base pair formation. This is an original diagram drawn by us.

Regarding the manipulation of the protein and nucleic acid, differential precipitation is a fundamental process of purification [82,95,98,99,100]. Ethanol and acetone precipitation of protein and nucleic acid is a classic technology for fractionation and concentration. The mechanism of ethanol-induced precipitation is often ascribed to the dehydration of bound water by proteins or nucleic acids [76,77,78,79,101,102,103]. However, we believe that the repulsive interactions of organic solvents are the main cause of the enhanced aggregation/association of proteins and nucleic acids. Repulsive interactions, which are principally energetically (thermodynamically) unfavorable, decrease inversely with the surface area; namely, the surface area decreases from unfolded to folded and then to aggregated proteins (or nucleic acids). This means that organic solvents stabilize a more compact state, namely, in the order of unfolded < folded < aggregated, provided that only repulsive interaction is involved in the effects of organic solvents. This is of course not correct, as described earlier, when organic solvents interact favorably with non-polar/aromatic side chains and nucleobases. This mechanism for organic solvents is true also for any excluded (repulsive) co-solvents, e.g., polymers. Thus, organic solvents (particularly ethanol) have been used to bring the precipitation of native proteins and nucleic acids. With nucleic acid precipitation, the presence of counter ions to reduce charge repulsion between nucleic acids in the precipitates is essential for organic solvents (that increase electrostatic repulsion due to the reduced dielectric constant) to induce nucleic acid precipitation. This is the major difference in organic solvent precipitation between protein and nucleic acid. Due to abundant solvent-exposed charges, the precipitation of nucleic acids by organic solvent requires charge neutralization by ions.

#### 3.4.2. Denaturants

The mechanisms by which denaturants such as urea and guanidine hydrochloride (GdnHCl) unfold proteins and nucleic acids may provide insights into the fundamental forces that drive their folding [104]. The denaturants are used to study the conformational transitions and stability of proteins and nucleic acids [90,105]. Another important application of denaturants is to solubilize denatured proteins, such as those expressed as inclusion bodies [106,107,108]. The solubilized proteins are then refolded to generate functional proteins for research and pharmaceutical applications. The expression and refolding technologies played a crucial role in producing proteins and advances in the development of research reagents and biopharmaceutical products, which were nearly impossible before the invention of recombinant technology.

How do the denaturants interact with proteins and nucleic acids? The interactions of urea and GdnHCl with amino acid side chains and peptide backbone were reported in pioneering studies [109,110,111,112]. The glycine solubility either changed little or slightly decreased with an increasing concentration of urea and GdnHCl, meaning that the interactions are neutral (no preference for polar glycine or water) or slightly repulsive between glycine and these denaturants and hence are not thermodynamically favorable. Namely, there appeared to be no affinity of the denaturants for dipolar glycine, which is markedly different from the strong repulsive interactions of organic solvents with glycine. Regarding hydrophilic threonine and glutamine, the interactions are attractive, but marginally small. In contrast, urea and GdnHCl showed strong attractive interactions with aromatic groups, like organic solvents. Urea has been shown to form π–π stacking interactions with aromatic groups of proteins [113,114,115]. GdnHCl and urea also favorably interact with the peptide groups, showing binding equilibrium with peptide groups in proteins [109,110,116]. The solubility of peptide groups increased with increasing denaturant concentration, clearly demonstrating the attractive interactions of peptide bonds with these denaturants, again markedly different from the repulsive interactions of organic solvents with peptide groups [109,110]. The attractive interactions of the denaturants with the peptide bonds were ascribed to bivalent hydrogen bonds [109,110].

We are not aware of attractive (or repulsive) interactions of urea and GdnHCl with nucleobases or phosphate backbones in nucleic acids. However, like the interactions of urea with aromatic side chains of proteins, urea can also confer π–π stacking with nucleobases; namely, urea confers attractive interactions with nucleobases [117]. Thus, the denaturants can also have affinity for aromatic nucleobases that are buried in the helical structures of nucleic acids (Figure 4). In addition, those co-solvents that favorably interact with the peptide bonds might interact similarly with the phosphate backbones in nucleic acids. For example, urea and GdnHCl that bind to the peptide bonds by bivalent hydrogen bonds can bind to the phosphate groups similarly [109,118], although these denaturants can interact with the side chains of both proteins and nucleic acids. Urea can cause the denaturation of nucleic acids by virtue of its binding to nucleobases through hydrogen bonding and aromatic/hydrophobic interactions. These attractive denaturant interactions with peptide groups stabilize the exposed peptide groups, which, upon unfolding, can lead to extended structures, as shown in Figure 10. GdnHCl can stabilize the backbone charges of nucleic acids, different from the effects of organic solvents, and hence may not cause the fully extended structure of nucleic acid, as shown in Figure 11.

#### 3.4.3. Polymers

Polyethylene glycol is perhaps the most frequently used polymer for fractionation and purification of proteins. The principal mechanism of the effects of polymers on protein precipitation and binding to an inert surface is steric hindrance, namely, the exclusion of polymers from the macromolecular surface, called “excluded volume effects” [119,120,121,122,123], “depletion effects” [124] or “macromolecular crowding [125,126]”. The exclusion principle was derived experimentally [119,120,121,122,127] and theoretically [123,125,126,128]. No matter how the effects are called, the effects of polymers arise from their inertness to the macromolecular surface and the large molecular size. Because of its high aqueous solubility, the availability of different sizes and modifications and cost, PEG is widely used for different applications. However, PEG is not necessarily inert, an often-cited misconception of the “inertness of PEG”, against proteins and nucleic acids. PEG has been shown to reduce the surface tension of water, independent of the size (from the molecular weight of 200 to 1000), suggesting that its effects on water structure are determined by the PEG monomeric unit [80]. The degree of surface tension depression by the PEG is greater than the typical organic solvents of dimethyl sulfoxide and methyl-pentanediol [80]. Hirano et al. [129] studied the interaction of PEG 4000 and 20,000 with amino acids in detail and derived the thermodynamic parameters of PEG interaction with the amino acid side chains, leading to the conclusion that “PEG behaves like a weak organic solvent”, as indicated in the paper title, and thus a conclusion that PEG is not definitely inert. Interestingly, PEG 20,000 resulted in the phase separation of glycine at the saturation concentration of glycine, one phase containing only glycine (glycine-rich phase) and another phase containing both PEG 20,000 and glycine (PEG-rich phase) [129]. Namely, PEG is enriched, and glycine is depleted in the PEG-rich phase, while in the glycine-rich phase, PEG is depleted. To summarize this observation, glycine and PEG 20,000 repel each other. Whether this is due to the excluded volume effects of PEG 20,000 or another mechanism cannot be answered from this experiment only. However, it has been shown that organic solvents, including ethanol [74], dioxane [74], ethylene glycol [81,129] and DMSO [74] decrease the solubility of glycine, indicating the unfavorable interaction of glycine with these organic solvents. Thus, it is possible that PEG is acting on glycine as an organic solvent. In fact, Hirano et al. [129] showed that PEG 400 and PEG 4000 equally decrease the glycine solubility, even more effectively than ethylene glycol on the weight concentration basis. The interaction energy of amino acid side chains with PEG was calculated from the solubility difference between glycine and amino acids, which requires the solubility data of glycine. Because of the phase separation of aqueous glycine solution into PEG-rich and glycine-rich phases with PEG 20,000, the interaction energy of side chains could not be obtained for PEG 20,000 (which requires the solubility data of glycine in PEG 20,000). Nevertheless, PEG 20,000 as well as PEG 4000 showed highly favorable interaction with tryptophan amino acid. This interaction is expected to have the potential contribution of unfavorable (repulsive) interaction of terminal amino and carboxyl groups of tryptophan (blue circle), depicted in Figure 12, as speculated from the repulsive interaction of PEG 20,000 and glycine. The favorable (attractive) interaction between PEG and indole structure (red circle) is expected to overwhelm the unfavorable contribution of overall excluded volume effect and charge–PEG interactions (blue circle). In fact, when the unfavorable contribution of the terminal interactions was subtracted, those side chains of not only tryptophan but also phenylalanine, leucine and isoleucine showed a favorable interaction with PEG 4000. This calculation was possible for PEG 4000, which caused no phase separation of glycine and allowed the determination of the glycine solubility measurement. This observation that PEG 4000 caused no phase separation is interesting. If organic solvent-like behavior is the sole factor, this PEG may be expected to cause phase separation. Thus, the reasons for the phase separation of PEG 20,000 may be both the organic solvent-like property and the excluded volume effect, the former being identical for PEG 4000 and 20,000 and the latter differing between these two PEGs.

This organic solvent-like property and excluded volume effect operate on the effects of PEG on protein and nucleic acid stability. PEG has been shown to decrease the stability of proteins to a very limited extent, which was explained by a balance between the excluded volume effect and PEG binding [121]. Such PEG binding clearly indicates that hydrophobic groups are buried inside the protein molecule and stabilized by hydrophobic PEG upon protein unfolding by heating. PEG, particularly small PEGs, resulted in a significant reduction in the melting of DNA [130], which can be explained by the hypothetical hydrophobic binding of PEG to the melted DNA and which in turn means that PEG binding requires the exposure of non-polar nucleobases. On the other hand, PEG binding to protein occurs to the hydrophobic side chains buried in the tertiary structure folding, not requiring the melting of the helical structure of the protein. A large PEG 8000 slightly increased the melting temperature (stabilized) of a 30-mer DNA duplex [131], indicating that a large-size PEG resulted in overwhelming the excluded volume effects on a large DNA duplex and meaning the “inertness” of such a large PEG on the large DNA.

Mode of PEG interaction with Trp terminal groups (blue circle) and side chain (red circle). This is an original diagram drawn by us.

A similar effect of PEG was observed with the quadruplex DNA helix. As depicted in Figure 13, there is small stabilization of the quadruplex structure by the PEG [130,131,132]. For the quadruplex helix, the excluded volume effect of PEG should have overwhelmed the aromatic–hydrophobic interactions between PEG and nucleobases, leading to stabilization of the quadruplex structure. The excluded volume effect of PEG on the quadruplex stabilization is schematically depicted in Figure 13, where the quadruplex structure is shown as a box (left) and its unfolded structure is shown as a rectangular box (right). PEG depicted as an ellipsoid is expected to be excluded to a greater extent for the unfolded DNA with a larger surface area than the folded quadruplex with reduced surface area, which would stabilize the folded structure. Alternatively, a dehydration mechanism was proposed as a major mechanism of the stabilization of the quadruplex structure. This mechanism is proposed in a classic work of Kauzmann as the hydrophobic hydration of a non-polar surface, the exposure of which is unstable in aqueous environments and reduced upon association with the non-polar surface [133]. This mechanism is equivalent to the excluded volume effects, which lead to loss of excess water upon the DNA folding, which arises from the exclusion of polymeric PEG. However, it should be noted that hydrophobic PEG may bind to the unfolded DNA by a hydrophobic interaction between the exposed nucleobases and the hydrophobic moiety of the PEG.

While the aromatic/hydrophobic interaction of PEG with nucleobases destabilize the helix, the excluded volume effects stabilize the helix. The hydration of nucleic acid is due to the excluded volume effects and decreases with the formation of a quadruplex structure from a single strand structure. This is an original diagram drawn by us.

## 4. Conclusions

We here described the similarities and differences in structure and interactions between proteins and nucleic acids. Their similarity in molecular architecture, i.e., backbone and side chains, leads to one common structure, a helix. Among the side chains, both protein and nucleic acid have aromatic groups, which play a central role in interactions. A major difference is their location. While the aromatic side chains of protein are excluded from participating in secondary structures, nucleobases participate in the helix formation in nucleic acid. Thus, the side chains in proteins are exposed to conferring molecular interactions for folding or protein–protein and protein–nucleic acid interactions, while those of nucleic acids require their exposure by strand breaks, loops, hairpins or kinks. This difference may be simply due to their physical location or different chemical properties. Both aromatic groups are similar in terms of hydrophobic properties in the sense that they interact favorably (attractively) with organic solvents and in terms of the aromatic nature in the sense that they participate in π–π interaction and cation (or anion)–π interaction. Particularly, they show favorable interaction with the arginine side chain. These interactions are thermodynamic parameters associated with the free energy changes in interactions and do not give any information on the physical interaction mechanism, e.g., the mechanism of aromatic–organic solvent interaction or aromatic–arginine interaction. In addition, we do not know the physical properties of aromatic–aromatic interaction between the side chains, which competes with aromatic–solvent interaction in the thermodynamic measurement. It would be of great interest to characterize this physical mechanism of aromatic–aromatic interaction and aromatic–solvent interaction side-by-side for nucleobases and amino acid side chains. The difference, if any, may give insight into why they play different roles in participating in secondary structure formation. Based on the knowledge obtained so far from co-solvent interactions, the aromatic properties of nucleic acid and protein side chains are similar, while their location in protein and nucleic acid structures is the main difference.

The structure of nucleopeptide may also give insight into the aromatic properties of nucleobases. When nucleobases are inserted into the peptide backbone, the structure does not appear to adopt a protein-type secondary structure. Instead, it forms base pairs with a complementary single strand nucleic acid, which may be due to the nature of the nucleobases. A question is whether a polypeptide comprising only aromatic amino acid side chains, e.g., polyphenylalanine or polytyrosine, would adopt a protein-type secondary structure or instead form aromatic stacking interactions that generate a different secondary structure.

Co-solvent interactions were used to draw an understanding of the properties of macromolecular interactions. It should be noted that co-solvents are used to stabilize proteins and nucleic acids or suppress their aggregations in pharmaceutical formulation development. Sucrose and trehalose are frequently used to enhance protein stability. Arginine has been shown to suppress protein aggregation during storage. Detergents are used to suppress surface-mediated aggregation. Increasingly, AI-based approaches, along with molecular simulations and docking, offer powerful tools to probe these interaction mechanisms and to guide rational co-solvent selection.

The folding of each protein and nucleic acid is well-known based on their similar architecture. However, this review focused on differences in the chemical natures of their backbone and side chains and how those differences lead to different interactions with other macromolecules and co-solvents. Such a comparison should give a basic understanding of the structural and functional properties of proteins and nucleic acids.

## Figures and Tables

**Figure 1 cimb-47-01019-f001:**
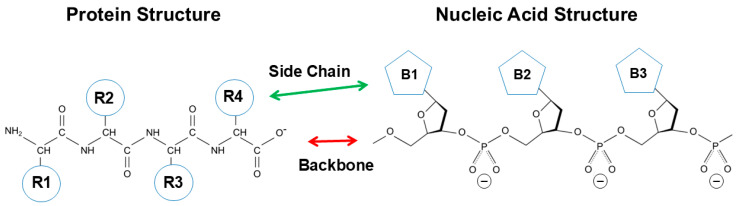
Comparison of protein and nucleic acid structures.

**Figure 2 cimb-47-01019-f002:**
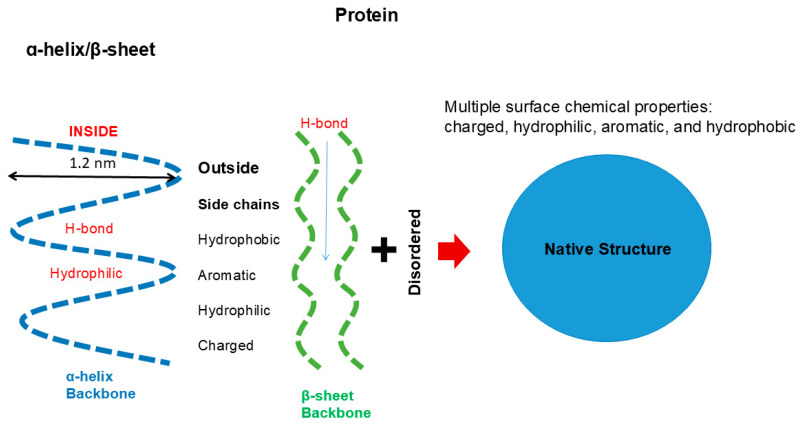
Secondary and tertiary structure of protein.

**Figure 3 cimb-47-01019-f003:**
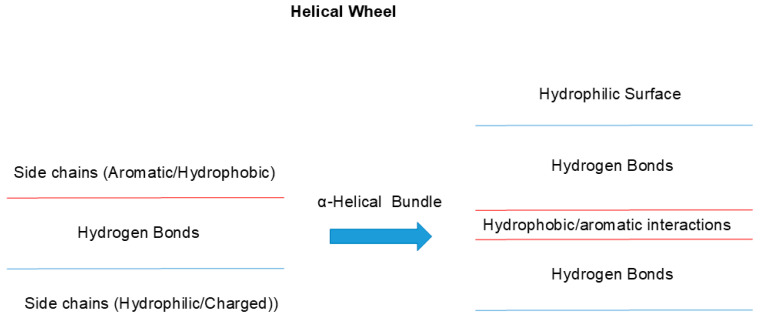
Architecture of helical wheels.

**Figure 4 cimb-47-01019-f004:**
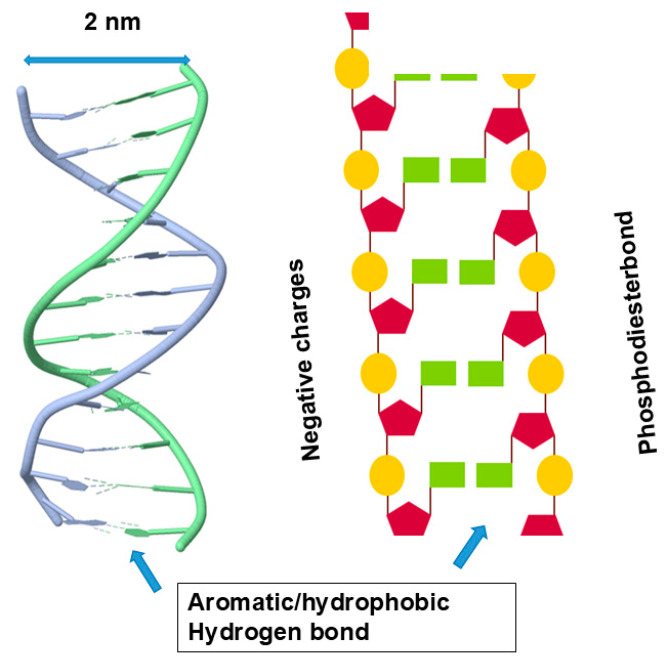
Secondary structure of double helix. The interior of the double helix is made of aromatic nucleobases, which would make the interior aromatic/hydrophobic. The outer surface has a constellation of negative charges arising from phosphate groups. Modified from https://search.aol.com/aol/search?q=nucleic+acid+structure+images&s_qt=ac&rp=&s_chn=prt_bon&s_it=comsearch (accessed on 1 December 2025).

**Figure 5 cimb-47-01019-f005:**
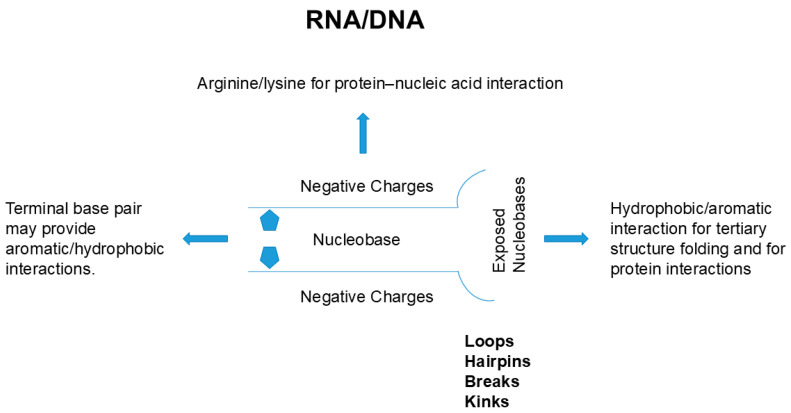
Schematic diagram of possible interactions between nucleic acids and proteins.

**Figure 6 cimb-47-01019-f006:**
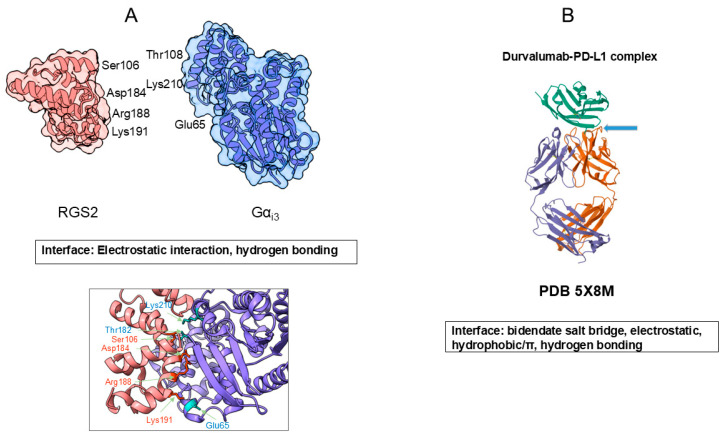
Model protein–protein interaction. (**A**). Interaction between human G-protein subunit alpha, G_αi3_ and engineered regulator of G-protein signaling type 2 domain, RGS2. (**B**). Antigen–antibody interaction between human antibody durvalumab Fab domain and antigen PD-L1. Structure A was rendered from PDB entry 2V4Z using UCSF ChimeraX version 1.10.1, and structure B was based on PDB ID 5 × 8 M.

**Figure 7 cimb-47-01019-f007:**
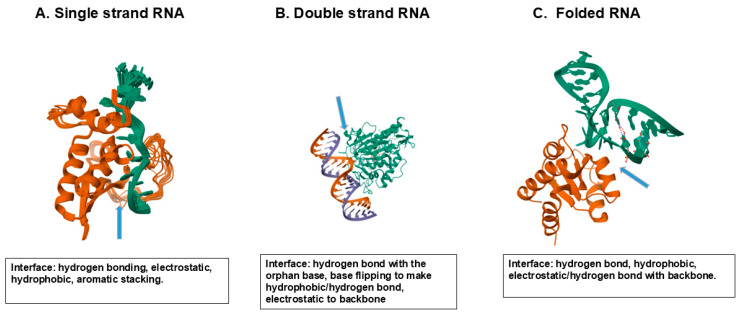
Model protein–nucleic acid interaction. (**A**) Protein binding to a single strand RNA. Splicing factor 1 (red). Single-stranded RNA (green). (**B**) Protein binding to a double helix RNA. ADAR2 deaminase (green). DNA double Helix (red/blue). (**C**) Protein binding to a folded RNA. L7Ae (red). K-turn RNA structure (green). Structures A, B and C are derived from PDB entries 1K1G, 5ED2 and 1RLG.

**Figure 8 cimb-47-01019-f008:**
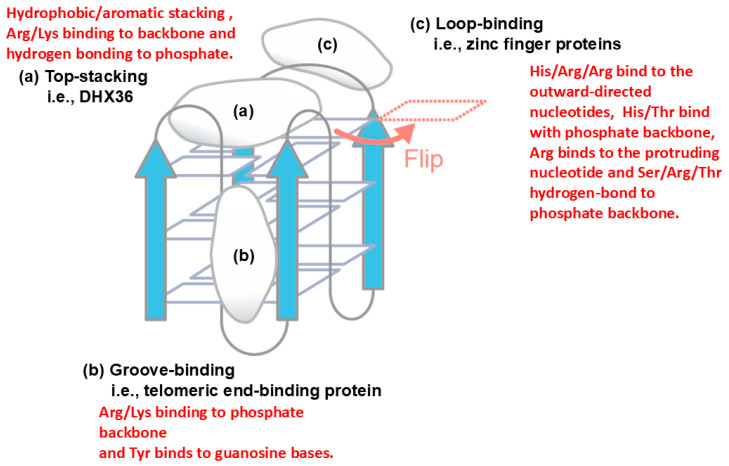
Potential interaction modes of a model quadruplex structure. (**a**) Top-stacking binding mode of DHX36. (**b**) Groove-binding mode of telomeric end-binding protein. (**c**) Loop-binding mode of zinc finger protein. The illustration was originally created with reference to Ref. [43].

**Figure 9 cimb-47-01019-f009:**
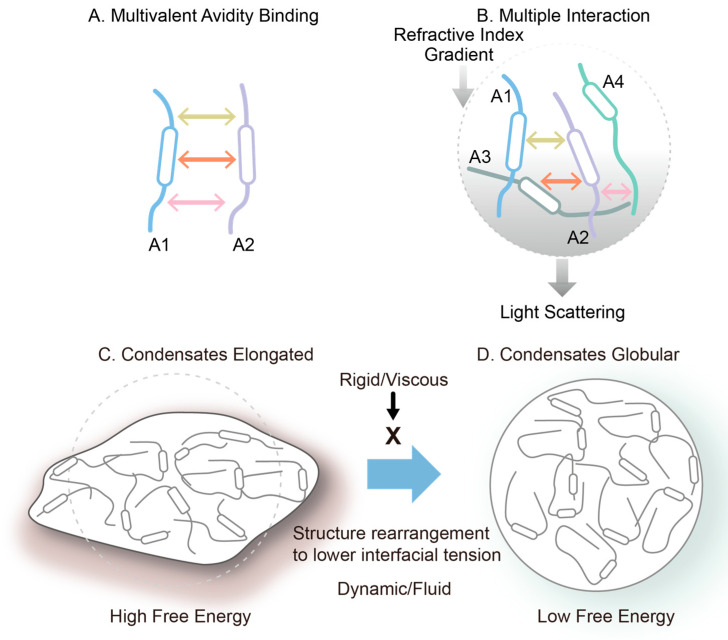
Schematic illustration of macromolecular assembly and condensate formation.

**Figure 10 cimb-47-01019-f010:**
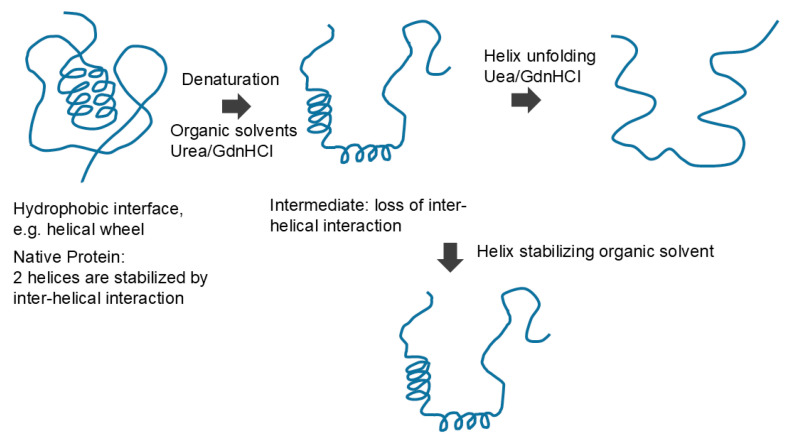
Schematic diagram of protein denaturation.

**Figure 11 cimb-47-01019-f011:**
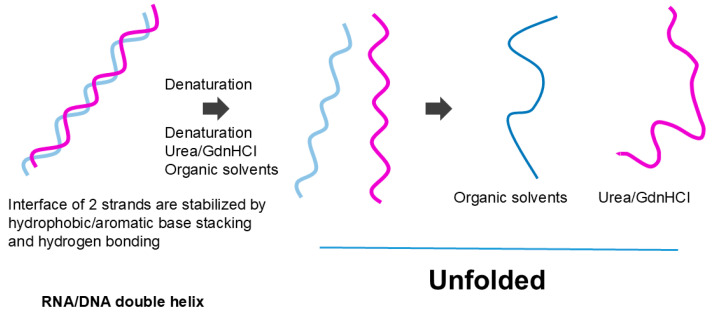
Unfolding of nucleic acid helix.

**Figure 12 cimb-47-01019-f012:**
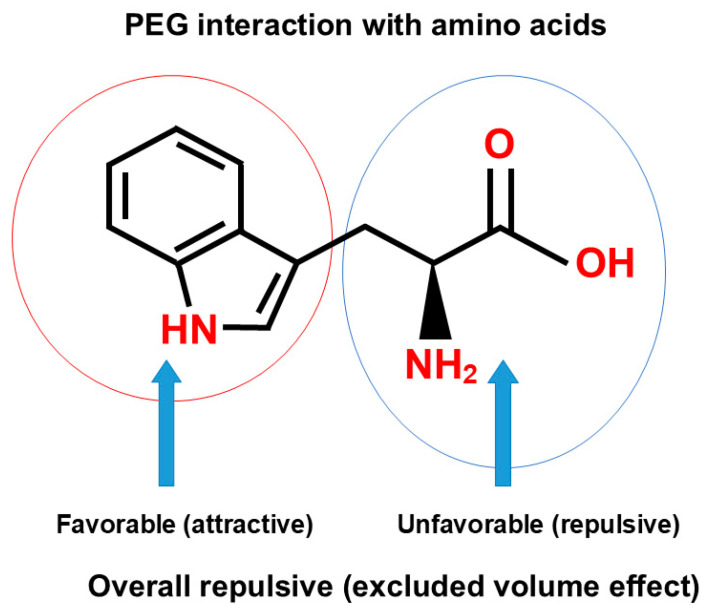
Schematic diagram of polymer interaction with tryptophan.

**Figure 13 cimb-47-01019-f013:**
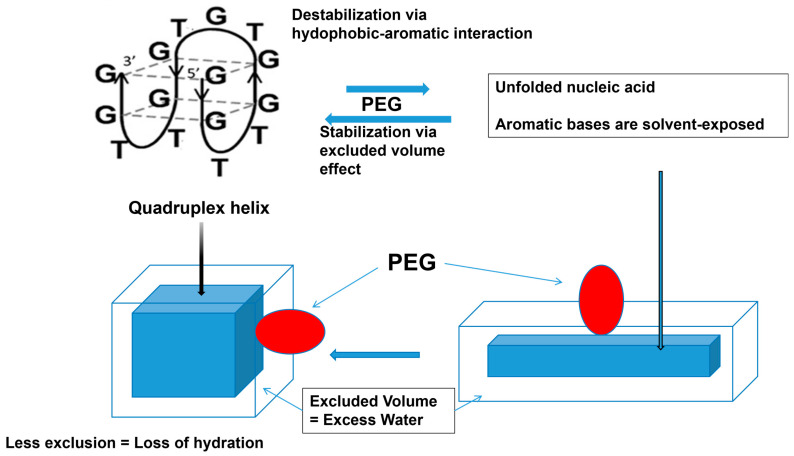
Schematic diagram of PEG interaction with quadruplex.

## Data Availability

No new data were created or analyzed in this study. Data sharing is not applicable to this article.

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
