# Peer review of "Differences and Similarities in Protein and Nucleic Acid Structures and Their Biological Interactions"

_cimb, 2025, doi:10.3390/cimb47121019_

Round 1
Reviewer 1 Report
Comments and Suggestions for Authors
The review article by Arakawa et al., offers a clear and structured comparison between the structural architectures of proteins and nucleic acids, emphasizing their backbones, side chains, folding patterns, and biological interactions. It extends the discussion to how these features influence interactions with co-solvents, which is relevant to both fundamental biology and biopharmaceutical applications. The article synthesizes several decades of biophysical literature into a cohesive narrative, and it includes numerous useful schematic figures illustrating secondary structures, macromolecular interactions, and condensate formation. Overall, the article is informative and could serve as a useful educational resource for students, early-career researchers, or those bridging biochemistry and biophysics.
The topic is appropriate for this journal, and the comparison of protein vs. nucleic acid physicochemical behavior is timely given emerging interest in biomolecular condensates and nucleopeptide engineering. However, the manuscript is quite long, contains repetition, and would significantly benefit from some re-organization, condensation, strengthened citations, and correction of grammatical errors. I recommend minor revisions before publication.
Major Comments
- The bibliography is rich but includes many older citations. More recent works (such as molecular simulations, docking, CRISPR- related technologies, and RNA therapeutics) could enhance the timeliness of the review.
- Some concepts, such as the description of covalent and non-covalent interactions are reiterated (almost verbatim). The manuscript can be condensed at least 30% more by reducing the redundancy.
- At certain places, specifically the abstract and the introduction, the manuscript is difficult to follow due to awkward phrasing and overly long sentences. Considering splitting the concepts in shorter sentences to improve readability.
- The later sections in the manuscript are descriptive, rather than analytical and quantitative. For example, the claims about PEG in the co-solvent interactions need more effective dissection to distinguish experimental evidence and author interpretation.
- The authors can clearly explain how the mechanistic comparison in this review article is novel and advances our current understanding beyond currently available literature.
Minor Comments
- Avoid first person sentences like “we believe”
- Back-bone should be backbone.
There are some very long sentences in the abstract and introduction, that impede readability. These can be improved by spliting into shorter sentences.
Author Response
Response to reviewer’s comments (cimb-3992692)
Reviewer #1
The review article by Arakawa et al., offers a clear and structured comparison between the structural architectures of proteins and nucleic acids, emphasizing their backbones, side chains, folding patterns, and biological interactions. It extends the discussion to how these features influence interactions with co-solvents, which is relevant to both fundamental biology and biopharmaceutical applications. The article synthesizes several decades of biophysical literature into a cohesive narrative, and it includes numerous useful schematic figures illustrating secondary structures, macromolecular interactions, and condensate formation. Overall, the article is informative and could serve as a useful educational resource for students, early-career researchers, or those bridging biochemistry and biophysics.
The topic is appropriate for this journal, and the comparison of protein vs. nucleic acid physicochemical behavior is timely given emerging interest in biomolecular condensates and nucleopeptide engineering. However, the manuscript is quite long, contains repetition, and would significantly benefit from some re-organization, condensation, strengthened citations, and correction of grammatical errors. I recommend minor revisions before publication.
Thank you for favorable comments. We have revised accordingly.
Major Comments
- The bibliography is rich but includes many older citations. More recent works (such as molecular simulations, docking, CRISPR- related technologies, and RNA therapeutics) could enhance the timeliness of the review.
Added more recent references and references to those on recommended topics.
2. Some concepts, such as the description of covalent and non-covalent interactions are reiterated (almost verbatim). The manuscript can be condensed at least 30% more by reducing the redundancy.
Tried to reduce or replace the word “Interactions”. However, we believe that
while they may be redundant in some places, they have significance in their
respective places, so decided to retain them as they are. We hope that would
make easier to evaluate the revised paper.
3. At certain places, specifically the abstract and the introduction, the manuscript is difficult to follow due to awkward phrasing and overly long sentences. Considering splitting the concepts in shorter sentences to improve readability.
Revised accordingly.
4. The later sections in the manuscript are descriptive, rather than analytical and quantitative. For example, the claims about PEG in the co-solvent interactions need more effective dissection to distinguish experimental evidence and author interpretation.
Revised accordingly.
5. The authors can clearly explain how the mechanistic comparison in this review article is novel and advances our current understanding beyond currently available literature.
Added one paragraph in conclusion about the focus of this review.
Minor Comments
- Avoid first person sentences like “we believe”
We felt simpler by starting a sentence with “We”.
2. Back-bone should be backbone.
We used backbone throughout.
Reviewer 2 Report
Comments and Suggestions for Authors
In review by T. Arakawa et al. ”Differences and similarities in protein and nucleic acid structures and their biological interactions” authors based on similarity and differences in structure of proteins and nucleic acids tried to explain its different interactions with denaturants, organic solvent and polymers.
Comments and suggestions:
- In my opinion, the authors should create a separate subsection in the manuscript on the role of arginine in protein-protein and protein-nucleic acid interactions. This will improve the perception of information.
- The most of figures (Figures 2, 3, 4, 5, 10, 11, 12, 13) are superfluous in the review. The presented schemes are rather primitive. There is an overlay of text in the Fig.9. On Figure 6B depicted complex of antigen with of Fab fragment of durvalumab not with full antibody as indicated in the figure caption.
- The effect of acetone on proteins and nucleic acids should be described also.
The English should be improved to more clearly express the manuscript. The review could be accepted after major revision.
Author Response
Reviewer #2
In review by T. Arakawa et al. ”Differences and similarities in protein and nucleic acid structures and their biological interactions” authors based on similarity and differences in structure of proteins and nucleic acids tried to explain its different interactions with denaturants, organic solvent and polymers.
Comments and suggestions:
- In my opinion, the authors should create a separate subsection in the manuscript on the role of arginine in protein-protein and protein-nucleic acid interactions. This will improve the perception of information.
Converted to a new section.
2. The most of figures (Figures 2, 3, 4, 5, 10, 11, 12, 13) are superfluous in the review. The presented schemes are rather primitive. There is an overlay of text in the Fig.9. On Figure 6B depicted complex of antigen with of Fab fragment of durvalumab not with full antibody as indicated in the figure caption.
Did not change Fig.2, 3, 4, 5 10, 11, 12 and 13, as we considered them
appropriate and necessary. For example, we feel that Fig.2 is necessary to see
hydrophilic nature of the interior of protein secondary structures and for the
subsequent discussion on differences in the interior of the nucleic acid helix.
Revised or corrected other figures accordingly.
3. The effect of acetone on proteins and nucleic acids should be described also.
Added.
The English should be improved to more clearly express the manuscript. The review could be accepted after major revision.
Inquiry from the journal office
- Reference 3 and 7
Replaced by corrected papers.
- Copy right
These figures have been modified and given sources.
Round 2
Reviewer 2 Report
Comments and Suggestions for Authors
Accept in present form